# One-shot Neural Backdoor Erasing via Adversarial Weight Masking

**Shuwen Chai** *
Renmin University of China
chaishuwen@ruc.edu.cn

**Jinghui Chen** †
Pennsylvania State University
jzc5917@psu.edu

## Abstract

Recent studies show that despite achieving high accuracy on a number of real-world applications, deep neural networks (DNNs) can be backdoored: by injecting triggered data samples into the training dataset, the adversary can mislead the trained model into classifying any test data to the target class as long as the trigger pattern is presented. Various methods have been proposed to nullify such backdoor threats. Notably, a line of research aims to purify the potentially compromised model. However, one major limitation of this line of work is the requirement to access sufficient original training data: the purifying performance is much worse when the available training data is limited. In this work, we propose Adversarial Weight Masking (AWM), a novel method capable of erasing the neural backdoors even in the one-shot setting. The key idea behind our method is to formulate this into a min-max optimization problem: first, adversarially recover the trigger patterns and then (soft) mask the network weights that are sensitive to the recovered patterns. Comprehensive evaluations of several benchmark datasets suggest that AWM can largely improve the purifying effects over other state-of-the-art methods on various available training dataset sizes.

## 1 Introduction

Deep neural networks (DNNs) have been widely applied in a variety of critical applications, such as image classification [17], object detection [48, 63] , natural language processing [9], and speech recognition [19], with tremendous success. The training of modern DNN models usually relies on large amount of training data and computation, therefore, it is common to collect data over the Internet or directly use pretrained models from third-party platforms. However, this also gives room for potential training-time attacks [43, 11, 21, 38, 40]. Particularly, backdoor attack [15, 32, 6, 43, 1, 33, 37, 45, 30] is among one of the biggest threats to the safety of the current DNN models: the adversary could inject triggered data samples into the training dataset and cause the learned DNN model to misclassify any test data to the target class as long as the trigger pattern is presented. In the meantime, the model still enjoy decent performances on clean tasks thus the backdoors can be hard to notice. Recent advanced backdoor attacks also adopt invisible [27], or even sample-specific [29] triggers to make it even stealthier.

Facing the immediate threat from backdoor adversaries, many backdoor defense or detection methods [31, 35, 16, 52, 58, 60] have been proposed. Particularly, we focus on a line of research which aims to purifying the potentially compromised model without any access to the model's training process. This is actually a quite realistic setting as the large-scale machine learning model nowadays [9, 2] can hardly be trained by individuals. Earlier works in this line usually purify the backdoored model via model fine-tuning [54, 7] or distillation [28, 14]. The problem is fine-tuning and distillation procedure

---

*This work was done when the author was remotely interned at Pennsylvania State University.
†Corresponding author.

can still preserve certain information on the backdoor triggers and thus it is hard to completely remove the backdoor. Moreover, since it is hard to for one to access the entire training data, longer time of fine-tuning of a small subset of data usually leads to overfitting and deteriorated model performances on clean tasks. In order to remove the backdoor in a more robust way, recent researches focus on removing the backdoor with adversarial perturbations [57, 60]. Particularly, [57] aims to extract sensitive neurons (by adversarial perturbations) that are highly related to the embedded triggers and prune them out. However, one major limitation is that it still requires to access sufficient original training data in order to accurately locate those sensitive neurons: the purifying performance is a lot worse when the available training data is insufficient. This largely limit the practicality of the defense as it can be hard to access sufficient original training data in real-world scenarios.

In this paper, we propose the Adversarial Weight Masking (AWM) method, a novel backdoor removal method that is capable of erasing the neuron backdoor even in the one-shot setting. Specifically, AWM adopts a minimax formulation to adversarially (soft) mask certain parameter weights in the neuron network. Intuitively, AWM aims to lower the weights on parameters that are related to the backdoor triggers while focusing more on the robust features [23]. Extensive experiments on backdoor removal with various available training data sizes demonstrate that our method is more robust to the available data size and even works under the extreme one-shot learning case while other baseline cannot. As a side product, we also found that AWM's backdoor removal performance for smaller sized networks are significantly better compared to other baselines.

## 2   Related Works

There exists a large body of literature on neural backdoors. In this section, we only review and summarize the most relevant works in backdoor attacks, defenses and adversarial training.

**Backdoor Attacks** The backdoor attack aims to embed predefined triggers into a DNN during training time. The adversary usually poisons a small fraction of training data through attaching a predefined trigger and relabeling them as corresponding target labels, which can be the same for all poisoned samples [6, 15] or different for each class [37]. In contrast, clean-label attacks [43, 1] only attach the predefined trigger to data from a target class and do not relabel any instances. On the design of backdoor triggers, BadNets attack [15] is the first to patch instances with a white square and reveal the backdoor threat in the training of DNNs. [32] optimizes trojan triggers by inversing the neurons. To make the triggers harder for detection, [45] proposed an adaptive adversarial training algorithm that maximizes the indistinguishability of the hidden representations of poisoned data and clean data while training. [30, 37] composites multiple or sample-aware trojan triggers to elude backdoor scanners. [6] first proposed the necessity of making triggers invisible and generated poisoned images by blending the backdoor trigger with benign images instead of by patching directly. Following this idea, some other invisible attacks [27, 29] are also prevailing, suggesting that poisoned images should be indistinguishable compared with their benign counter-part to evade human detection.

**Backdoor Defenses** Opposite to backdoor attack, backdoor defense aims to *detect a triggered model* or *remove the embedded backdoor*. For the purpose of detection, the defender may detect abnormal data before model training [47, 35, 10, 12] or identify poisoned model after training [53, 59]. Another line of research focuses on backdoor removal through various techniques including fine-tuning [52, 16, 54], distillation [7], or model ensemble [28, 25]. DeepSweep [41] searches data augmentation functions to transform the infected model as well as the inference samples to rectify the model output of trigger-patched samples. However, this method relies on the access to the poisoned data. Recently, [60] formalizes backdoor removal as a minimax problem and utilizes the implicit hypergradient to solve it. As it needs fine-tuning the parameters, performance decay may happen when the available fune-tuning data is limited. Another latest work [57] discovers that backdoored DNNs tend to collapse and predict target label on clean data when neurons are perturbed, and therefore pruning sensitive neurons can purify the model. From empirical studies, we still discover that it cannot maintain its efficacy with a small network and one-shot learning.

**Adversarial Training** Our work is also related to study of adversarial training [36], which adopts minmax robust optimization techniques for defending against adversarial examples [13, 50, 22, 5, 4, 8]. [61] theoretically studies the trade-off between natural accuracy and robust accuracy. [62] proposes friendly adversarial training with better trade-off between natural generalization for adversarial

robustness. Recent study [56] also reveals the relationship between robustness and model width. Several works also study accelerating adversarial training in practice [44, 3, 55].

# 3 Preliminaries and Insignts

## 3.1 Preliminaries

**Defense Setting.** We adopt a typical defense setting where the defender outsourced a backdoored model from an untrusted adversary. The defender is not aware of whether the model is been backdoored or which is the target class. The defender is assumed to have access to a small set of training data (or data from the same distribution) but no access to the entire original training data.

**Adversarial Neuron Pruning.** ANP [57] is one of the state-of-the-art backdoor removal method that adversarially perturbs and prunes the neurons without knowing the exact trigger patterns.

Denote $\mathbf{w}$ and $\mathbf{b}$ as the weight and bias of the network. Considering a DNN $f$ with $L$ layers, let's denote the $k$-th neuron in the $l$-th layer as $z_k^{(l)} = \sigma(\mathbf{w}_k^{(l)}\mathbf{z}^{(l-1)} + b_k^{(l)})$, where $\sigma$ is the activation function. ANP works by first finding the neurons that are possibly compromised to the trigger patterns and then prune them out to remove the backdoors. Specifically, it will first perturb all the neurons in DNN by multiplying small numbers $\delta_k^{(l)}$ and $\xi_k^{(l)}$ on the corresponding weight $\mathbf{w}_k^{(1)}$ and bias $\mathbf{b}_k^{(1)}$ respectively. Then we have $z_k^{(l)} = \sigma((1+\delta_k^{(l)})\mathbf{w}_k^{(l)}\mathbf{z}^{(l-1)} + (1+\xi_k^{(l)})\mathbf{b}_k^{(l)})$ as the new neuron output. To simplify the notation, let's denote $\circ$ as the above multiplication on the neuron-level, $n$ as the total number of neurons, $\epsilon$ the maximum level of perturbation. Then the goal of this perturbation is to find the perturbation that can maximize the classification loss:

$$\max_{\boldsymbol{\delta},\boldsymbol{\xi}\in[-\epsilon,\epsilon]^n} \mathbb{E}_{(\mathbf{x},y)\sim D}\, \mathcal{L}(f(\mathbf{x};(1+\boldsymbol{\delta})\circ\mathbf{w},(1+\boldsymbol{\xi})\circ\mathbf{b}),y) \tag{3.1}$$

Note that $\boldsymbol{\delta}$ and $\mathbf{w}$ have different dimensions so that the perturbation is not weight-wise but neuron-wise. Those weights corresponding to the same neuron are multiplied with the same perturb fraction $\delta$. ANP claimed that by solving problem (3.1), we can identify sensitive neurons related to potential backdoors. With the solved $\boldsymbol{\delta}$ and $\boldsymbol{\xi}$, the second step is to optimize the mask for neurons with the following objective:

$$\min_{\mathbf{m}\in\{0,1\}^n} \mathbb{E}_{(\mathbf{x},y)\sim D}\, \alpha\mathcal{L}(f(\mathbf{x};\mathbf{m}\circ\mathbf{w},\mathbf{b}),y) + \beta \max_{\boldsymbol{\delta},\boldsymbol{\xi}\in[-\epsilon,\epsilon]^n} \mathcal{L}(f(\mathbf{x};(\mathbf{m}+\boldsymbol{\delta})\circ\mathbf{w},(1+\boldsymbol{\xi})\circ\mathbf{b}),y)$$
$$\tag{3.2}$$

By solving the above min-max optimization, the poisoned model prunes those sensitive neurons detected by neuron perturbation and removes the potential backdoors. Note that when BatchNorm (BN) [24] layer is used, ANP's perturbation on $\mathbf{w}$ and $\mathbf{b}$ will be canceled out by the batch normalization and nothing changes after BN layers. Therefore, the implementation ANP directly perturb the scale and shift parameters in the BN layers in such cases.

## 3.2 Problems of ANP

ANP [57] claims to be an effective backdoor removal method without knowing the exact trigger pattern, and since it does not really fine-tune the model but directly prune the neurons, it can preserve decent model accuracy on the clean tasks. However, its backdoor removal performance largely depends on the effectiveness of identifying the sensitive neurons regarding the backdoor trigger: if Eq. (3.2) failed to identify the accurate binary mask $\mathbf{m}$, ANP will perform badly on backdoor removal tasks. Unfortunately, in certain practical settings, ANP does fail to: 1) remove the backdoor when the available clean training data size is extremely small; 2) maintain high accuracy on clean tasks when the network size is small and the BN layer is used.

We select the BadNets attack for illustration and set the target class as 8 to train the backdoored models. First, we test the ANP performances with various sizes of available training data. The left part of Figure 1(a) shows that the perturbed neurons (by ANP) tend to predict the target class a lot more often than other classes when the size of available training data is sufficient, however, when the size of available data drops to 10, it can no longer effectively indicates such pattern and the prediction portion on different classes distributes quite evenly. As an immediate result, ANP's backdoor removal performance significantly degrades when the size of available data is small (Figure 1(a) right part).

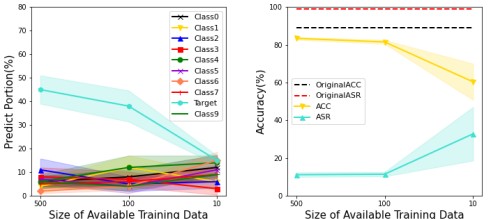 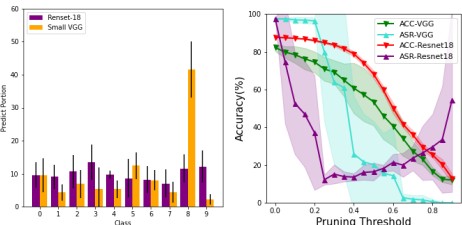

(a) The left shows the prediction portion for each class with perturbed neurons with various available training data size. The right shows the ASR/ACC of the ANP pruned models with various available training data size.

(b) The left shows the prediction portion for each class with perturbed neurons with ResNet and VGG model. The right shows the ASR/ACC of the ANP pruned models under various pruning threshold.

Figure 1: An illustrative example of the failure cases of ANP.

We then investigate how the network size affects ANP's performance by applying it on both VGG (small) and ResNet-18 (large) backdoored models. The left part of Figure 1 (b) indicates that while ANP's perturb neuron is able to show larger prediction portion on the target class, when applying on smaller VGG model, its magic failed again. The right part of Figure 1 (b) illustrates the ASR/ACC of the ANP pruned models under various pruning threshold. We can observe that it is hard to find a suitable pruning threshold for the smaller VGG network to obtain both high ASR and low ACC.

## 4 Our Proposed Method

In this section, we introduce our proposed method. Inspired by the above analysis in Section 3, we propose Adversarial Weight Masking (AWM) for better backdoor removal under practical settings.

**Soft Weight Masking.** From the analysis in Section 3, the neuron pruning method can be inappropriate when the backdoored model (with BN layers) is small and only has few layers: pruning certain neurons in the BN layer cuts off the information from a whole channel, which inevitably ignores some certain beneficial information for the clean tasks. To fix this drawbacks, we propose to adopt weight masking instead of neuron pruning:

$$\min_{\mathbf{m}\in[0,1]^d} \mathbb{E}_{(\mathbf{x},y)\sim D}\ \alpha\mathcal{L}(f(\mathbf{x};\mathbf{m}\odot\boldsymbol{\theta}),y) + \beta\max_{\boldsymbol{\delta}\in[-\epsilon,\epsilon]^d}\mathcal{L}(f(\mathbf{x};(\mathbf{m}+\boldsymbol{\delta})\odot\boldsymbol{\theta}),y),\qquad(4.1)$$

where $\boldsymbol{\theta}\in\mathbb{R}^d$ denotes the entire neural network weights, $\boldsymbol{\delta}$ denotes the small perturbations on the network parameters, $\mathbf{m}$ is the weight mask of the same dimension as $\boldsymbol{\theta}$, $\odot$ denotes the Hadamard product (element-wise product). Eq. (4.1) follows the general idea of ANP by first identifying the sensitive part of the neural network and then lower such sensitivity. The major changes here is that we are no longer pruning out the neurons, instead, we add an additional mask for all the network weights. Note that such design would provide more flexibility in removing backdoor-related parts and thus avoid over-killing in BN layers. Since we apply weight masking instead of neuron pruning, we can also use soft mask $\mathbf{m}\in[0,1]^d$ instead of binary neuron masks as in ANP [57].

**Adversarial Trigger Recovery.** Another issue identified in Section 3 is that ANP performs poorly when the available training data size is small. And it seems that under such challenging conditions, perturbing the mask itself does not give clues to which part of the network is really sensitive to the backdoor triggers. Inspired from adversarial training literature [36], we can first optimize the following objective for adversarially estimating the worst-case trigger patterns:

$$\max_{\|\boldsymbol{\Delta}\|_1\leqslant\tau} \mathbb{E}_{(\mathbf{x},y)\sim D}\ \mathcal{L}(f(\mathbf{x}+\boldsymbol{\Delta};\boldsymbol{\theta}),y),\qquad(4.2)$$

where $\|\cdot\|_1$ denotes the $L_1$ norm and $\tau$ limits the strength of the perturbation. Note that technically speaking, Eq. (4.2) only aims to find a $L_1$ norm universal perturbation that can mislead the current model toward misclassification. Yet since we are not aware of the target class, this is a reasonable surrogate task for the trigger recovery. Based on Eq. (4.2), we can integrate it with soft weight masking and formulate it as a min-max optimization problem:

$$\min_{\mathbf{m}\in[0,1]^d} \mathbb{E}_{(\mathbf{x},y)\sim D}\ \alpha\mathcal{L}(f(\mathbf{x};\mathbf{m}\odot\boldsymbol{\theta}),y) + \beta\max_{\|\boldsymbol{\Delta}\|_1\leqslant\tau}[\mathcal{L}(f(\mathbf{x}+\boldsymbol{\Delta};\mathbf{m}\odot\boldsymbol{\theta}),y)],\qquad(4.3)$$

where $\alpha$ and $\beta$ are tunable hyper-parameters.

**Sparsity Regularization.** To push our defense to mask out backdoor-related weights more aggressively, we adopt the $L_1$ norm regularization on $\mathbf{m}$ for further controlling its sparsity level, as inspired by previous works [52, 49] on trigger optimization.

Combining soft weight masking, adversarial trigger recovery together with sparsity regularization on $\mathbf{m}$, gives the full *Adversarial Weight Masking* formulation:

$$\min_{\mathbf{m} \in [0,1]^d} \mathbb{E}_{(\mathbf{x},y) \sim D} \, \alpha \mathcal{L}(f(\mathbf{x}; \mathbf{m} \odot \boldsymbol{\theta}), y) + \beta \max_{\|\boldsymbol{\Delta}\|_1 \leqslant \tau} [\mathcal{L}(f(\mathbf{x} + \boldsymbol{\Delta}; \mathbf{m} \odot \boldsymbol{\theta}), y)] + \gamma \|\mathbf{m}\|_1, \quad (4.4)$$

where $\alpha$, $\beta$ and $\gamma$ are tunable hyper-parameters. Intuitively, AWM works by first identifying the worst-case universal triggers (which are highly likely to be the actual triggers or different patterns with similar backdoor effects), and then finding an optimal weight mask $\mathbf{m}$ to lower the importance on the identified triggers while maintaining the accuracy on clean tasks.

Unlike ANP, which directly prunes out the suspicious neurons, we aim at learning a soft mask for each parameter weight, i.e., each element in $\mathbf{m}$ lies in between $[0, 1]$. Such design can help preserve the information beneficial to the clean tasks and thus avoid over-killing. Moreover, adopting soft masks can also avoid the problem of setting the hyper-parameters on the pruning threshold, which is also heuristic and hard to generalize for various experimental settings.

---

**Algorithm 1** Adversarial Weight Masking (AWM)

---

**Input:** Infected DNN $f$ with $\boldsymbol{\theta}$, Clean dataset $D = \{(\mathbf{x}_i, y_i)\}_{i=1}^n$, Batch size $b$, Learning rate $\eta_1, \eta_2$, Hyper-parameters $\alpha, \beta, \gamma$, Epochs $E$, Inner iteration loops $T$, $L_1$ norm bound $\tau$.

1: Initialize all elements in $\mathbf{m}$ as 1
2: **for** $i = 1$ to $E$ **do**
3:     Initialize $\boldsymbol{\Delta}$ as $\mathbf{0}$
    *// Phase 1: Inner Optimization*
4:     **for** $t = 1$ to $T$ **do**
5:         Sample a minibatch $(\mathbf{x}, y)$ from $D$ with size $b$
6:         $\mathcal{L}_{inner} = \mathcal{L}(f(\mathbf{x} + \boldsymbol{\Delta}; \mathbf{m} \odot \boldsymbol{\theta}), y)$
7:         $\boldsymbol{\Delta} = \boldsymbol{\Delta} - \eta_1 \nabla_{\boldsymbol{\Delta}} \mathcal{L}_{inner}$
8:     **end for**
9:     Clip $\boldsymbol{\Delta}$: $\boldsymbol{\Delta} = \boldsymbol{\Delta} \times \min(1, \frac{\tau}{\|\boldsymbol{\Delta}\|_1})$
    *// Phase 2: Outer Optimization*
10:    **for** $t = 1$ to $T$ **do**
11:       $\mathcal{L}_{outer} = \alpha \mathcal{L}(f(\mathbf{x}; \mathbf{m} \odot \boldsymbol{\theta}), y) + \beta \mathcal{L}(f(\mathbf{x} + \boldsymbol{\Delta}; \mathbf{m} \odot \boldsymbol{\theta}), y) + \gamma \|\mathbf{m}\|_1$
12:       $\mathbf{m} = \mathbf{m} + \eta_2 \nabla_{\mathbf{m}} \mathcal{L}_{outer}$
13:       Clip $\mathbf{m}$ to $[0, 1]$.
14:    **end for**
15: **end for**
    **Output:** Filter masks $\mathbf{m}$ for weights in network $f$.

---

**Algorithm Details.** The detailed steps of AWM is summarized in Algorithm 1. We solve the min-max optimization problem in Eq. (4.4) by alternatively solving the inner and outer objectives. Specifically, we initialize all the mask values as 1. In each epoch, we repeat the following steps: 1) initialize $\boldsymbol{\Delta}$ as $\mathbf{0}$, and then perform $K$-steps of gradient descent on $\boldsymbol{\Delta}$ and clip it with its $L_1$ norm limit $\tau$; 2) we update soft weight mask in the outer optimization via stochastic gradient descent, where the first term is to minimize the clean classification loss, the second term is for lowering the weights associated with $\boldsymbol{\Delta}$, and the third term is the $L_1$ regularization on $\mathbf{m}$, followed by a clipping operation to keep $\mathbf{m}$ within $[0, 1]$. Note that we reinitialize $\boldsymbol{\Delta}$ in each inner optimization as we need to relearn the adversarial perturbation based on the current $\mathbf{m} \odot \boldsymbol{\theta}$. We also on purposely set $T > 1$ for ensuring sufficient optimization during each update in order to reach better convergence.

## 5 Experiments

In this section, we conduct thorough experiments to verify the effectiveness of our proposed AWM method and analyze the sensitivity on hyper-parameters via ablation studies.

**Datasets and Networks.** We conduct experiments on two datasets: CIFAR-10 [26] and GTSRB [20]. CIFAR-10 contains 50000 training data and 10000 test data of 10 classes. GTSRB is a dataset of traffic signal images, which contains 39209 training data and 12630 test data of 43 classes. The poisoned model is trained with full training data with $5\%$ poison rate on Resnet-18 [18], ShuffleNet V2 [34], MobileNet V2 [42], or a small VGG [46] network with three simplified blocks, containing six convolution layers followed by BN layers. See Appendix A for more experimental setting details and Appendix B for results on GTSRB.

**Attacks and Defenses.** For the backdoor attack baselines, we consider 1) *BadNets with square trigger (BadNets)* [15]; 2) *Trojan-Watermark (WM)* and *Trojan-Square (SQ)* [32]; 3) $l_0$-*inv* and $l_2$-*inv* [27], two invisible attack methods with different optimization constraints; 4) *Blended attack (BLD)* [6]; 5) *Clean-label attack (CLB)* [51]; 6) *WaNet* [38] with both single target and *all-to-all (a2a)* target settings. We mainly compare our method with two latest state-of-the-art methods of backdoor removal: *Implicit Backdoor Adversarial Unlearning (IBAU)* [60], *Adversarial Neuron Pruning (ANP)* [57].

**Evaluations.** We adopt two metrics: ACC and ASR. ACC is the test accuracy on clean dataset, while ASR is calculated as the ratio of those triggered samples that are still predicted as the adversary's target labels. Note that usually a benign classifier is not assciated with a specific trigger, thus its prediction on poisoned data mainly follows its prediction on clean data. Under such case, suppose we have $c$ classes in total, we can expect the ASR should be around $1/c$, that is, $10\%$ for CIFAR-10 and $2.3\%$ for GTSRB. Therefore, once the backdoor removal method achieves an ASR close to $1/c$ (less than $1.5/c$), we consider it as successfully remove the backdoor (rather than achieving ASR$= 0\%$).

## 5.1 Backdoor Removal with Various Available Data Size

Table 1: Backdoor removal performance comparison with various available data sizes on CIFAR-10 dataset with Resnet-18 and VGG Net. Numbers represent percentages. **Bold** numbers indicate the best ACC after backdoor removal and blue numbers indicate successful backdoor removal.

| Attack | Available Data Size n | Resnet-18 Origin | ANP ACC | ANP ASR | IBAU ACC | IBAU ASR | AWM(Ours) ACC | AWM(Ours) ASR | VGG Net Origin | ANP ACC | ANP ASR | IBAU ACC | IBAU ASR | AWM(Ours) ACC | AWM(Ours) ASR |
|---|---|---|---|---|---|---|---|---|---|---|---|---|---|---|---|
| BadNets | 5000 | ACC | 85.56 | 10.18 | 86.41 | 11.26 | **86.94** | 10.46 | ACC | 77.34 | 8.64 | 81.06 | 12.25 | **83.58** | 13.98 |
| | 500 | 87.83 | 83.39 | 11.15 | 84.88 | 35.61 | **83.56** | 12.11 | 85.98 | 73.17 | 13.76 | 77.30 | 13.52 | **78.20** | 11.93 |
| | 200 | ASR | 83.52 | 11.53 | 82.38 | 83.89 | **84.26** | 10.90 | ASR | 64.59 | 13.35 | 75.88 | 14.75 | **76.42** | 12.82 |
| | 100 | 97.90 | 81.48 | 11.42 | 78.80 | 97.82 | **83.57** | 11.10 | 97.96 | 51.19 | 15.86 | 75.53 | 33.62 | **75.69** | 10.64 |
| | 50 | | **81.09** | 11.21 | 73.84 | 98.92 | 80.46 | 11.42 | | 49.81 | 17.66 | 68.72 | 45.23 | **73.20** | 12.22 |
| Trojan-SQ | 5000 | ACC | **87.30** | 10.66 | 86.34 | 9.38 | 87.08 | 11.21 | ACC | 67.70 | 8.68 | 82.38 | 14.20 | **83.82** | 12.76 |
| | 500 | 88.27 | 85.34 | 9.34 | 81.08 | 10.38 | **86.30** | 10.34 | 85.86 | 63.21 | 35.77 | 76.42 | 11.53 | **79.40** | 10.08 |
| | 200 | ASR | 82.72 | 10.51 | 75.72 | 99.94 | **85.38** | 9.41 | ASR | 63.84 | 36.31 | 73.81 | 10.69 | **75.50** | 14.40 |
| | 100 | 99.61 | 80.28 | 7.42 | 66.38 | 93.82 | **85.68** | 10.32 | 99.36 | 40.23 | 7.14 | 74.32 | 55.68 | **74.49** | 12.08 |
| | 50 | | 69.68 | 9.29 | 39.83 | 98.80 | **80.78** | 8.48 | | 40.06 | 6.41 | 73.20 | 84.32 | **72.23** | 5.01 |
| Trojan-WM | 5000 | ACC | 85.72 | 38.48 | 84.68 | 14.32 | **87.12** | 12.92 | ACC | 58.14 | 31.70 | **83.03** | 8.26 | 82.78 | 13.64 |
| | 500 | 88.00 | 82.82 | 34.06 | 80.63 | 10.22 | **85.17** | 8.36 | 86.08 | 55.64 | 9.76 | **82.89** | 7.33 | 82.61 | 12.15 |
| | 200 | ASR | 83.43 | 66.30 | 80.32 | 20.68 | **84.88** | 11.10 | ASR | 52.58 | 8.45 | 80.27 | 10.36 | **81.96** | 17.88 |
| | 100 | 99.96 | 75.99 | 61.64 | 78.75 | 38.82 | **83.31** | 12.51 | 99.80 | 42.95 | 21.20 | 81.02 | 30.25 | **81.56** | 12.82 |
| | 50 | | 70.52 | 9.33 | 69.42 | 99.78 | **80.14** | 3.43 | | 46.84 | 6.15 | 78.33 | 35.06 | **79.97** | 8.88 |
| $l_0$ inv | 5000 | ACC | 86.08 | 15.20 | 85.32 | 10.72 | **86.38** | 11.74 | ACC | 66.90 | 10.21 | **82.90** | 12.68 | 82.26 | 12.88 |
| | 500 | 88.23 | 83.71 | 15.08 | 80.83 | 14.48 | **84.97** | 11.81 | 86.56 | 67.70 | 30.20 | **80.42** | 10.11 | 75.01 | 20.54 |
| | 200 | ASR | **83.47** | 18.18 | 75.83 | 28.90 | 82.83 | 17.79 | ASR | 69.47 | 73.10 | **76.26** | 95.50 | 76.20 | 33.74 |
| | 100 | 100.0 | 77.32 | 16.44 | 73.49 | 70.18 | **82.04** | 12.68 | 100.0 | 60.31 | 59.14 | **67.40** | 93.56 | 62.31 | 24.58 |
| | 50 | | 69.21 | 25.26 | 69.83 | 85.34 | **77.68** | 25.73 | | 54.95 | 58.08 | 59.13 | 78.20 | **60.73** | 45.36 |
| $l_2$ inv | 5000 | ACC | 85.04 | 12.14 | 86.46 | 7.28 | **87.22** | 10.76 | ACC | 70.70 | 7.58 | 81.51 | 6.23 | **82.74** | 12.94 |
| | 500 | 88.51 | 82.25 | 31.99 | 78.66 | 9.32 | **85.76** | 10.26 | 86.22 | 74.80 | 0.44 | 78.09 | 7.64 | **81.33** | 4.39 |
| | 200 | ASR | 82.21 | 30.68 | 77.38 | 50.46 | **85.16** | 11.45 | ASR | 66.38 | 0.92 | 73.28 | 6.42 | **80.36** | 6.39 |
| | 100 | 99.86 | 81.80 | 21.68 | 73.26 | 90.48 | **82.26** | 8.85 | 99.84 | 53.07 | 1.12 | 72.91 | 18.86 | **81.67** | 7.55 |
| | 50 | | 72.65 | 8.90 | 63.21 | 93.46 | **75.60** | 10.86 | | 47.87 | 0.15 | 75.41 | 30.27 | **80.36** | 9.93 |

We first study the backdoor removal performances of AWM on various available data sizes and compare with other state-of-the-art defense baselines. Table 1 presents the defense results on the CIFAR-10 dataset. Specifically, among the entire CIFAR-10 training data, 2500 images are backdoored. We test with varying size of available data samples ranging from 5000 to 10 for each defense. A fixed number of 5000 remaining samples are used to evaluate the defense result.

The left column depicts five single-target attack methods, and the first row represents two different adopted network structures. Note that each attack's poison rates are set to be $5\%$. We present the ACC and ASR under each backdoor removal setting in the table. All single-target attacks are capable of achieving an ASR close to $100\%$ and an ACC around $88\%$ with no defenses. For Resnet-18, the performance of the baselines are comparable with AWM when there are sufficient available training data ($n = 5000$): all methods effectively remove the backdoors. With the decreasing size of clean data, IBAU suffers from huge performance degradation and fails to remove the backdoor under several settings. The major reason is that its fine-tuning procedure can actually hurt the original information stored in the parameters that are crucial to its clean accuracy, especially when fine-tuning on small sample set. On the other hand, ANP shows better robustness as it prunes the neurons which reduces the negative effect of insufficient data, but still fails on more challenging cases. On the right part of Table 1, we can observe that ANP losses more accuracy on the small VGG network, which backup our analysis in Section 3. AWM shows state-of-the-art backdoor removal performances with various available data sizes, network structures and successfully erase the neuron backdoors in most cases.

Table 2: An Extreme Case: One-Shot Backdoor Removal Comparison on CIFAR-10 Data. Numbers represent percentages. **Bold** numbers indicate the best ACC after backdoor removal and blue numbers indicate successful backdoor removal.

| Method | BadNets | | Trojan-SQ | | Trojan-WM | | $l_0$ inv | | $l_2$ inv | |
|---|---|---|---|---|---|---|---|---|---|---|
| | ACC | ASR | ACC | ASR | ACC | ASR | ACC | ASR | ACC | ASR |
| Origin | 87.83 | 97.90 | 88.27 | 99.61 | 88.00 | 99.96 | 88.23 | 100.0 | 88.51 | 99.86 |
| ANP | 60.35 | 32.83 | 68.32 | 13.88 | 50.42 | 35.50 | 63.42 | 22.46 | 67.08 | 76.16 |
| IBAU | 60.18 | 97.33 | 45.38 | 96.27 | 57.76 | 99.93 | 69.26 | 95.81 | 63.48 | 89.42 |
| **AWM (Ours)** | **76.46** | 8.98 | **78.26** | 10.68 | **74.28** | 8.66 | **69.94** | 10.18 | **76.60** | 10.64 |

We further conduct experiments in an extreme one-shot setting, i.e., we only provide one image per class as the available data for backdoor removal tasks (total size as 10 for CIFAR-10 dataset). Table 2 shows the result of ACC and ASR under such a one-shot setting. In this case, we randomly sample one image for each of the ten classes and use the basic data-augmentation method, such as random horizontal flip and random crop. As a result, our AWM successfully removes all those backdoors with minimal performance drop ($10\%$ higher than other baselines on average). In contrast, other baselines failed in removing the existing backdoor triggers for most cases (as suggested by the large ASR values).

## 5.2 Reliability of AWM under Different Attack Types and Network Architectures

To comprehensively show our performance on different network structures and attack types, we select two more lightweight networks and four more backdoor attacks to conduct experiments under the one-shot setting. We summarize the ACC and ASR on CIFAR-10 in Table 3.

**Robustness on Network Architectures.** Different from the VGG we adopted to illustrate "a small network", ShuffleNet and MobileNet contain more convolutional and BN layers while their parameters are fewer.

Table 3: Backdoor removal performance comparison with lightweight architectures and universal or all-to-all attacks.

| | ShuffleNet V2 | | | | | | | |
|---|---|---|---|---|---|---|---|---|
| | BLEND | | CLB | | WaNet | | WaNet(a2a) | |
| | ACC | ASR | ACC | ASR | ACC | ASR | ACC | ASR |
| No Defense | 84.37 | 99.93 | 83.41 | 99.78 | 89.85 | 99.22 | 89.57 | 84.25 |
| ANP | 44.15 | 32.51 | 62.47 | 6.53 | 64.39 | 4.77 | 72.58 | 10.03 |
| AWM | **69.75** | 2.69 | **70.33** | 2.19 | **76.35** | 7.49 | **75.81** | 9.73 |
| | MobileNet V2 | | | | | | | |
| No | 88.83 | 99.78 | 87.70 | 100.0 | 93.78 | 91.01 | 94.08 | 92.72 |
| ANP | 51.95 | 85.33 | 57.65 | 22.09 | 75.31 | 18.97 | **79.28** | 10.33 |
| AWM | **67.87** | 2.10 | **66.68** | 9.70 | **74.29** | 8.92 | 80.97 | 13.38 |

parameters are fewer. Although the performance gap between ANP and AWM closes up a little bit, AWM still consistently cleans up all the backdoors with better clean task performances. This phenomenon also gives us insights that the performance of neuron pruning methods may be related to network depth more than the amount of parameters. To sum up, AWM works well on various structures of small neural networks with limited resources.

**Robustness on Trigger Types.** AWM accommodates different types of triggers as it does not make an assumption on the formulation of triggers. Although we used the adversarial trigger recovery and sparsity regularization as parts of techniques, they do not put a constraint on the triggers to be fixed or sparse but help remove the non-robust features in the poisoned network from the root.

The results on Blend, CLB, and WaNet also support the reasoning. The Blend attack uses gaussian noise (poison rate $= 1\%$) that covers the whole image. We adopted the CLB attack with adversarial perturbations and triggers on four corners. WaNet warps the image with a determined warping field as poisoned data, so there does not exist a fixed trigger pattern. As it uses a noisy warping field to generate noisy data with actual labels, it is difficult to train a backdoored model with a poison rate of $1\%$. We set the poison rate as $10\%$. These three attacks cover scenarios that triggers are dynamic and natural. Thus the experimental results verify that AWM does not rely on universal and sparse triggers. The above-mentioned attacks are mostly all-to-one, except for the CLB. To fill the blank of all-to-all attacks, we perform all-to-all attacks with WaNet on CIFAR-10 and the all-to-all attack with Trojan-SQ pattern on GTSRB in Appendix B. In conclusion, AWM performs well on fixed, universal, dynamic triggers and all clean label, all-to-one, and all-to-all attack settings.

## 5.3 Ablation Study on Each Component of AWM

Table 4: The Effect of Each Component: From ANP to AWM. $+$ and $-$ indicate an increase or decrease in accuracy. $\downarrow$ indicates large improvements in lowering ASR. R denotes Resnet-18.

| Attack&Network | Avail. Data Size | ANP ACC | ANP ASR | ANP+SWM ACC | ANP+SWM ASR | ANP+SWM+ATR ACC | ANP+SWM+ATR ASR | Full AWM ACC | Full AWM ASR |
|---|---|---|---|---|---|---|---|---|---|
| BadNets (R) | 500 | 83.39 | 11.15 | 83.25 (-0.14) | 12.62 | 84.78 (+1.53) | 12.08 | 85.33 (+0.55) | 11.76 |
| | 100 | 81.48 | 11.42 | 82.25 (+0.77) | 13.04 | 83.83 (+1.58) | 9.55 | 83.57 (-0.26) | 11.10 |
| | 10 | 53.26 | 34.38 | 73.34 (+19.9) | 10.16 ↓ | 80.38 (+7.04) | 10.41 | 76.46 (-3.92) | 8.98 |
| Trojan-SQ (R) | 500 | 85.34 | 9.34 | 85.27 (-0.07) | 12.00 | 84.06 (-1.19) | 9.02 | 84.91 (-0.85) | 10.20 |
| | 100 | 80.28 | 7.42 | 82.01 (+1.73) | 10.35 | 83.23 (+1.22) | 11.95 | 85.07 (+1.84) | 11.34 |
| | 10 | 68.32 | 13.88 | 73.12 (+4.80) | 11.42 | 82.04 (+8.92) | 10.72 | 78.26 (-3.78) | 10.68 |
| Trojan-WM (R) | 500 | 82.82 | 34.06 | 83.07 (+0.25) | 9.34 ↓ | 85.23 (+2.16) | 7.79 | 84.88 (-0.35) | 10.12 |
| | 100 | 75.99 | 31.64 | 78.23 (+2.24) | 15.02 ↓ | 82.99 (+4.76) | 4.61 ↓ | 84.21 (+1.22) | 11.18 |
| | 10 | 50.42 | 35.50 | 61.64 (+11.2) | 17.88 ↓ | 75.66 (+14.0) | 7.54 ↓ | 74.28 (-1.38) | 8.66 |
| $l_0$ inv (R) | 500 | 83.71 | 15.08 | 83.31 (-0.40) | 11.67 | 84.14 (+0.83) | 13.91 | 84.83 (0.69) | 12.15 |
| | 100 | 77.32 | 16.44 | 81.16 (+3.84) | 13.11 | 84.39 (+3.23) | 17.87 | 82.44 (+1.95) | 11.97 |
| | 10 | 63.42 | 22.46 | 65.46 (+2.04) | 10.40 | 73.66 (+8.20) | 14.70 | 69.94 (+3.72) | 10.18 |
| $l_2$ inv (R) | 500 | 82.25 | 31.99 | 82.59 (+0.34) | 13.94 ↓ | 82.15 (-0.45) | 6.26 | 85.22 (+3.07) | 13.13 |
| | 100 | 81.80 | 21.68 | 80.51 (-1.29) | 10.47 ↓ | 81.08 (+0.57) | 11.24 | 79.79 (+1.29) | 11.77 |
| | 10 | 67.08 | 76.16 | 60.36 (-6.72) | 12.20 ↓ | 66.78 (+6.42) | 15.80 | 76.60 (+9.82) | 10.64 |
| $l_2$ inv (VGG) | 500 | 74.80 | 0.44 | 76.35 (+1.55) | 3.17 | 82.08 (+5.73) | 5.81 | 81.33 (-0.75) | 4.39 |
| | 100 | 66.38 | 0.92 | 75.63 (+9.25) | 7.89 | 79.42 (+3.79) | 6.46 | 80.36 (+0.94) | 6.39 |
| | 10 | 47.08 | 30.15 | 70.82 (+23.7) | 19.17 | 78.34 (+7.52) | 14.73 | 80.32 (+1.98) | 12.52 |

We further perform an ablation study on each component of AWM. For notational simplicity, we refer the soft weight masking as SWM, and adversarial trigger recovery as ATR. From left to right in Table 4, we demonstrate the performance of the original ANP method, ANP + SWM (as in Eq. (4.1)), ANP + SWM + ATR (as in Eq. (4.3), and our full AWM method.

Table 4 shows that each component in AWM is non-trivial and necessary since adding each component would enhance the performance on average. Previous analysis in Section 3 suggests two of the ANP's weaknesses: when the network is small and when the available training data size is small. The first weakness motivates us to adopt soft label masking. As expected, SWM contributes more with the small VGG net and verifies that it overcomes the drawback of neuron pruning in a smaller network's BN layer. The second weakness motivates us to perform adversarial trigger recovery. From Table 4 we can easily observe ATR's improvements in lowering the ASR and significantly improving the ACC. The effect of $L_1$ regularization is comparably small but indeed forces the mask $\mathbf{m}$ to be sparser and thus further lowering the influence of weights associated with the recovered trigger patterns.

## 5.4 Additional Ablation Studies

In this section, we perform additional empirical studies on the necessity of regularization and AWM's robustness on the hyper-parameters. We compare our AWM with the following modified models: 1) *No Clip*: AWM with no $\Delta$ clipping; 2) *No Shrink*: AWM with no $L_1$ regularization on $\mathbf{m}$; 3) *NC-NS*: AWM with no $\Delta$ clipping and $\mathbf{m}$ regularization; 4) $L_2$ *Reg*: AWM with $\Delta$'s $L_2$ regularization; 5) $L_2$ *Reg NC*: AWM with $\Delta$'s $L_2$ regularization and no clipping.

Table 5: Ablation Study on AWM. $\downarrow$ indicates significant performance drop; $\uparrow$ indicates negative effect on backdoor removal. The base for comparison is Full AWM.

| Avail. Data Size | Method | BadNets ACC | BadNets ASR | Trojan-SQ ACC | Trojan-SQ ASR | Trojan-WM ACC | Trojan-WM ASR | $l_0$ inv ACC | $l_0$ inv ASR | $l_2$ inv ACC | $l_2$ inv ASR |
|---|---|---|---|---|---|---|---|---|---|---|---|
| 200 | No Clip | 82.86 $\downarrow$ | 19.36 $\uparrow$ | 79.21 $\downarrow$ | 20.58 $\uparrow$ | 84.82 | 32.16 $\uparrow$ | 80.17 $\downarrow$ | 46.85 $\uparrow$ | 81.76 $\downarrow$ | 17.28 $\uparrow$ |
| | No Shrink | 84.52 | 10.31 | 83.06 $\downarrow$ | 9.20 | 84.33 | 9.96 | 83.34 | 16.52 | 84.66 | 10.43 |
| | NC-NS | 82.33 $\downarrow$ | 15.78 $\uparrow$ | 78.41 $\downarrow$ | 26.93 $\uparrow$ | 84.40 | 37.81 $\uparrow$ | 77.84 $\downarrow$ | 36.37 $\uparrow$ | 81.80 $\downarrow$ | 12.39 |
| | $L_2$ Reg | 81.46 $\downarrow$ | 13.29 | 83.60 $\downarrow$ | 8.81 | 83.93 | 14.63 | 83.04 | 18.52 | 85.30 | 9.48 |
| | $L_2$ Reg NC | 83.72 | 11.64 | 83.49 $\downarrow$ | 30.13 $\uparrow$ | 83.55 $\downarrow$ | 7.56 | 81.27 $\downarrow$ | 29.61 $\uparrow$ | 83.45 $\downarrow$ | 21.54 $\uparrow$ |
| | Full AWM | 84.26 | 10.90 | 85.38 | 9.41 | 84.88 | 11.10 | 82.83 | 17.79 | 85.16 | 11.44 |
| one-shot | No Clip | 66.03 $\downarrow$ | 16.28 $\uparrow$ | 62.14 $\downarrow$ | 20.68 $\uparrow$ | 55.32 $\downarrow$ | 12.28 | 61.68 $\downarrow$ | 37.38 $\uparrow$ | 72.71 | 16.15 $\uparrow$ |
| | No Shrink | 66.32 $\downarrow$ | 9.97 | 76.62 | 12.83 | 73.50 | 9.26 | 70.69 | 21.36 $\uparrow$ | 75.57 | 14.86 |
| | NC NS | 65.32 $\downarrow$ | 9.77 | 68.17 $\downarrow$ | 26.14 $\uparrow$ | 73.62 | 59.52 $\uparrow$ | 70.22 | 24.87 $\uparrow$ | 71.52 | 29.84 $\uparrow$ |
| | $L_2$ Reg | 72.71 | 8.98 | 75.21 | 8.06 | 71.32 | 8.48 | 72.42 | 14.73 | 76.96 | 12.35 |
| | $L_2$ Reg NC | 73.96 | 14.38 | 72.50 $\downarrow$ | 13.74 | 73.39 | 10.61 | 68.94 | 31.53 $\uparrow$ | 72.06 | 20.87 $\uparrow$ |
| | Full AWM | 76.46 | 8.98 | 78.26 | 10.68 | 74.28 | 8.66 | 69.94 | 10.18 | 76.60 | 10.64 |

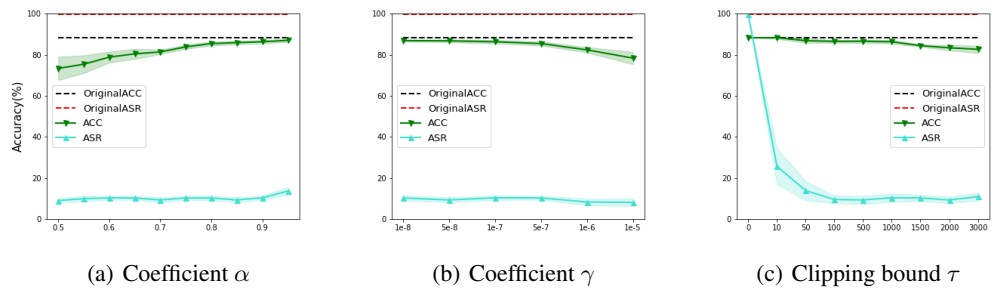

(a) Coefficient $\alpha$      (b) Coefficient $\gamma$      (c) Clipping bound $\tau$

Figure 2: Sensitivity on hyper-parameters. Performance ($\pm$std) over 5 random runs is reported.

**Constraints on $\Delta$ and $\mathbf{m}$.** In Table 5, we compare the results of different modifications of AWM. On the one hand, the clipping of the virtual trigger $\Delta$ is necessary as *No Clip* and $L_2$ *Reg NC* either remove the backdoor incompletely or sacrifice the accuracy too much. $L_2$ *Reg* changes the form of regularization and achieves comparable results on several settings but is less stable than AWM. The comparison between AWM and $L_2$ *Reg* also shows that both $L_1$ and $L_2$ norm regularization work for $\Delta$. On the other hand, the regularization of $\mathbf{m}$ helps better learn the soft mask. *NC-NS* differs from *No Clip* only in the $\mathbf{m}$ but successfully unlearns more backdoors. This is also reasonable since: by punishing the $L_1$ norm, the soft masks are forced to reach smaller values and thus be more aggressive on suspicious trigger-related features.

**Hyper-parameters.** We first use one single defend task to test AWM's sensitivity to hyper-parameters: the coefficient $\alpha$, $\gamma$, and the clipping bound $\tau$ for $\Delta$. Concretely, $\alpha$ controls the trade-off between the clean accuracy and backdoor removal effect, $\gamma$ controls the sparsity of weight masks, and $\tau$ controls the maximum $L_1$ norm of the recovered trigger pattern. We test with $\alpha \in [0.5, 0.8], \beta = 1 - \alpha, \gamma \in [10^{-8}, 10^{-5}], \tau \in [10, 3000]$ and shows the performance changes under the Trojan-SQ attack with 500 training data. When varying the value of one specific hyper-parameter, we fix the others to the default value as $\alpha_0 = 0.9, \gamma_0 = 10^{-7}, \tau_0 = 1000$. As shown in Figure 2, $\gamma$ is quite robust within the selected range. However, if we choose an overly large $\gamma$, the mask would shrink its value and hurt the accuracy. $\alpha$ works best around $0.8$ to $0.9$. If $\alpha$ is too close to 1, the major goal of AWM will shift to maintain clean accuracy while paying less attention to backdoor removal. The clipping bound $\tau$ should also be selected within a moderate range, as the adversarial perturbation should neither be too small to fail in capturing the real trigger nor be too large to lead to difficulties in finding the optimal

soft mask $\mathbf{m}$. In addition, considering the data dimension for CIFAR-10, further increasing $\tau$ after 3000 is the same as no constraint and thus meaningless.

As observed, the impacts of $\alpha$ and $\gamma$ on the performance has a single decrease or increase, while the clipping bound $\tau$ needs a moderate value. To verify the difficulty of selecting $\tau$, we then conduct experiments to compare the $\tau$s from 10 to 3000 for four attacks, keeping $\alpha$ and $\gamma$ to be 0.9 and $10^{-7}$.

Observing the ACC and ASR curves with Trojan-SQ, Trojan-WM, BadNets, and $L_0$-inv in Figure 3, the ACC decreases with the increase of $\tau$ while ASR first decreases fastly and then slightly increases again or keeps flat. The reasons behind this are easy to interpret. In terms of ACC, since we are using a larger $\tau$, the model weights are masked to adapt to triggered data points that are far different from the original data distribution. As a result, this will unavoidably hurt the original task ACC. On the other hand, in an ideal condition, ASR should decrease since the solution space of the optimization problem with larger $\tau$ actually contains the solution of the same problem with a smaller $\tau$. Thus in terms of backdoor removal, it should be at least as good as that of a smaller $\tau$. However, in practice, since we use PGD for solving the inner maximization problem, setting a large $\tau$ will simply make the $L_1$ norm of the recovered trigger too large. Thus the recovered trigger will be further away from the actual trigger and negatively affects the backdoor removal performances, especially on the variance.

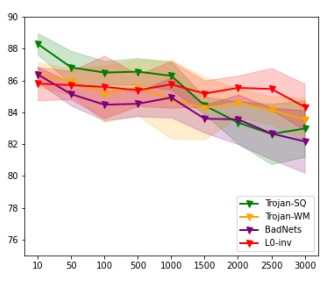

(a) ACC

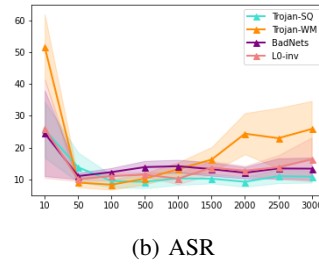

(b) ASR

Figure 3: Sensitivity on $\tau$ over different defend tasks. Performance ($\pm$std) over 5 random runs is reported.

Fortunately, the workable range of $\tau$ is indeed vast for various attacks, and it is pretty safe to select a value between 50 and 1000 for a $32 \times 32$ image, which can be easily extended to other image sizes by the ratio $\tau/$image size. On the one hand, even if we falsely select an overly-large $\tau$, such as 3000 for BadNets, the resulting ACC and ASR are still acceptable. On the other hand, we recommend selecting small $\tau$ because it tends to outperform the default setting. It is also reasonable to assume that the adversary tends to make a trigger invisible and small in the norm. For example, the $L_1$ norm of Trojan-WM's actual trigger is around 115, which is way smaller than our default value of $\tau = 1000$. In fact, $\tau = 1000$ roughly means that we allow the trigger to significantly change the clean image over $16 \times 16$ pixels in a $32 \times 32$ image which has been fairly large.

## 6    Conclusions and Future Works

In this work, we propose a novel Adversarial Weight Masking method which adversarially recovers the potential trigger patterns and then lower the parameter weights associated to the recovered patterns. One major advantage of our method is its ability to consistently erasing neuron backdoors even in the extreme one-shot settings while the current state-of-the-art defenses cannot. Under scenarios with a few hundred clean data, AWM still outperforms other backdoor removal baselines. Extensive empirical studies show that AWM relies less on the network structure and the available data size than neuron pruning based methods.

Note that currently, our AWM method still needs at least one image per class in order to erase the neuron backdoors properly. In some cases, the defender may have no clue what the training data is. Under such settings, it would be wonderful if the model owner could also remove the backdoors in a data-free setting. In addition, it is still under-explored how the weight masks in different layers contribute to the results. It would also be interesting to explore whether an attacker could break the backdoor defense method with the knowledge of the optimization details. We leave these problems as future works.

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
