# OpenReview forum: "One-shot Neural Backdoor Erasing via Adversarial Weight Masking"
_NeurIPS.cc/2022/Conference — NeurIPS 2022 Accept_

### Official Review · Reviewer_T69H · 2022-07-09

**Rating:** 5
**Confidence:** 4
**Soundness:** 2 fair
**Presentation:** 3 good
**Contribution:** 2 fair

**Summary:**

This paper try to address two main drawbacks of adversarial neuron pruning (ANP), that is, ANP 1) largely depends on the size of the training set and 2) performs badly when it fails to identify sensitive neurons. On this basis, the authors propose three strategies, namely soft weight masking (SWM), adversarial trigger recovery (ATR) and sparsity regularization (SR). The proposed methods outperform ANP especially when the available dataset is small and the network size is small.

**Questions:**

a. As the authors claim that the proposed methods help when the network size is small, they should test their methods on more typical lightweight architectures such as ShuffleNet [A], SqueezeNet [B] and MobileNet [C]. The results only on VGG is unconvincing.

b. The defense effect against stronger attacks (e.g., clean label attack [D], blended attack [E]) and under various poisoning rate (especially 1%) should be tested.

c. I wonder if the proposed defense work against all-to-all attacks in one-shot setting?

[A] Zhang, X., Zhou, X., Lin, M. and Sun, J., 2018. Shufflenet: An extremely efficient convolutional neural network for mobile devices. In Proceedings of the IEEE conference on computer vision and pattern recognition (pp. 6848-6856).

[B] Iandola, F.N., Han, S., Moskewicz, M.W., Ashraf, K., Dally, W.J. and Keutzer, K., 2016. SqueezeNet: AlexNet-level accuracy with 50x fewer parameters and< 0.5 MB model size. arXiv preprint arXiv:1602.07360.

[C] Howard, A.G., Zhu, M., Chen, B., Kalenichenko, D., Wang, W., Weyand, T., Andreetto, M. and Adam, H., MobileNets: Efficient Convolutional Neural Networks for Mobile Vision Applications.

[D] Turner, A., Tsipras, D. and Madry, A., 2019. Label-Consistent Backdoor Attacks.

[E] Chen, X., Liu, C., Li, B., Lu, K. and Song, D., Targeted Backdoor Attacks on Deep Learning Systems Using Data Poisoning.

**Ethics Review Area:**

["Privacy and Security (e.g., consent)"]

**Limitations:**

No. One of the limitations of the proposed methods is that it assumes the trigger is universal and sparse, which do not cover dynamic triggers [36, 37] and triggers that are spread over the whole image [33].

**Strengths And Weaknesses:**

Strengths

a. The paper is clearly written and easy to follow, and the motivation is reasonable.

b. The proposed methods outperform ANP in limited resources cases, especially in one-shot setting.

Weaknesses:

a. The evaluation is insufficient.

b. Although AWM seems better than ANP, the generalization to different attacks is considered not as good as ANP. This is because the authors assume the trigger is universal (ATR) and sparse (SR), which do not cover dynamic triggers [36, 37] and triggers that are spread over the whole image [33].

---

> ### Author Response · Authors · 2022-08-02
> **Response to Reviewer T69H**
>
> Thanks for your suggestions on additional experiments.
>
> Q1: Evaluate ANP with more small networks to valid its usability.
>
> A1: We additionally test and compare AWM with ANP on ShuffleNet[A] and MobileNet[C]. We summarize the ACC(ASR) on CIFAR10 in the following tables. The results show that our method still outperforms significantly better with limited resources. The VGG we adopted to illustrate "a small network" has fewer layers than ShuffleNet and MobileNet, while ShuffleNet and MobileNet have fewer parameters. To sum up, AWM works well on various structures of small neural networks. Note that we only report results on the one-shot unlearning case for illustration. With more training resources, all methods will perform better as shown in Table 1 in the paper and Table 3 in the appendix.
>
> | **ShuffleNetV2**  | **Blend**    | **CLB**      | **WaNet**    | **WaNet(a2a)** |
> | ----------------- | ------------ | ------------ | ------------ | -------------- |
> | Original ACC(ASR) | 84.37(99.93) | 83.41(99.78) | 89.85(99.22) | 89.57(84.25)   |
> | ANP               | 44.15(32.51) | 62.47(6.53)  | 64.39(4.77)  | 72.58(10.03)   |
> | AWM               | 69.75(2.69)  | 70.33(2.19)  | 76.35(7.49)  | 75.81(9.73)    |
>
>
>
> | **MobileNetV2**  | **Blend**    | **CLB**      | **WaNet**    | **WaNet(a2a)** |
> | ----------------- | ------------ | ------------ | ------------ | -------------- |
> | Original ACC(ASR) | 88.83(99.78) | 87.70(100.0) | 93.78(91.01) | 94.08(92.72)   |
> | ANP               | 51.95(85.33) | 57.65(22.09) | 75.31(18.97) | 79.28(10.33)   |
> | AWM               | 67.87(2.10)  | 66.68(9.70)  | 74.29(8.92)  | 80.97(13.38)   |
>
> Q2: ANP’s reliability on universal and sparse triggers.
>
> A2: We use the ATR and SR as components in our objective function, but we do not limit the triggers as universal and sparse. Our method can also work under other kinds of attacks. The additional results on Blended attack [E] (Blend), Clean-label attack [D] (CLB), and WaNet [36] also support the reasoning. Blend uses gaussian noise (poison rate = 0.01) that covers the whole image. We used CLB with adversarial perturbations and triggers on four corners. WaNet warps the image with a determined warping field as poisoned data, so there is not a fixed trigger pattern. Since it uses a noisy warping field to generate noisy data (with true labels), it is difficult to train a backdoored model with a poison rate of 0.01. We use poison rate = 0.10 and noise rate = 0.20. These three attacks cover scenarios that triggers are dynamic and natural. Our performance verifies that AWM does not rely on universal and sparse triggers.
>
> Q3: The availability of ANP under all-to-all attacks.
>
> A3: The above results include additional all-to-all(a2a) attacks with WaNet. We have presented results on the all-to-all attack with the gtsrb dataset in the supplementary materials in the original submission (see Appendix Section B Table 3). In conclusion, AWM performs well on both all-to-one and all-to-all attacks.

---

> > ### Comment · Reviewer_T69H · 2022-08-06
> > **Reply to the authors**
> >
> > Thanks for the authors' effort on the experiments. The results on small networks are acceptable to me. The performance of ANP and AWM on WaNet (with dynamic and global trigger) are much closer, which is as expected.
> >
> > Here I have another question. Since there are three hyperparameter that need to be tuned, I wonder if the authors use the same setting of hyperparameters against all attacks? Because in practice, we do not know which attacks will be used. It is important that we can use the same setting of hyperparameters that can defense most of the attacks.  The results will be of little significance if one should carefully adjust the hyperparameters for each attack. As far as I know, ANP can achieve better results if their hyperparameters are very carefully tuned.
> >
> > I'm glad to see that the authors study the sensitivity of the hyperparameter in section 5.3, but the results are shown only on one type of attack. It is unclear whether the hyperparameters still robust under other attacks. According to my experience, the sensitive of hyperparameters can be very different under different attacks.
> >
> >  In summary, there are two facts about the hyperparameters that should be considered, one is the robustness of the choice under each attack, which have been studied by the authors under only _one_ type of attack. Another one is the robustness of one setting against all the attacks. The latter one is more important to me.
> >
> > I will raise my score if my concern is addressed.

---

> > > ### Author Response · Authors · 2022-08-08
> > > **Response to Reviewer T69H**
> > >
> > > Thanks for your reply and suggestions on the two types of evaluations.
> > >
> > > First, we want to assure you that we used the same hyper-parameters for most of the attacks and our hyperparameters have a wide adaptability to different types of attacks. We totally agree with you that as defenders, we usually do not have much prior knowledge about the attack.
> > >
> > > Let us revisit the three hyperparameters in our method: $\alpha$ controls the trade-off between the clean accuracy and backdoor removal effect. $\gamma$ controls the sparsity of weight masks. $\tau$ controls the maximum $L_1$ norm of the recovered trigger pattern. In our experiments, we keep $\alpha$ and $\gamma$ to be 0.9 and $10^{-7}$ across all the attacks in our experiments. We slightly vary the choice of $\tau$. The best value of $\tau$ is indeed related to the attack method but only to a limited degree. A default value of $\tau=1000$ can also provide similar results. We previously set $\tau=2000$ for Badnets, $\tau=100$ for Trojan-WM, and $\tau=1000$ for other attacks (including a2a, blend, clb, and wanet). To showcase that this choice does not have a huge effect, we also re-do the Badnets and Trojan-WM experiments with the default value of $\tau=1000$. The following table shows that the performance gap is actually quite small
> > >
> > > | | Badnets  | Trojan |
> > > | -------- | ----- | ----- |
> > > | Fine-tuned | 83.56(12.11) |  85.17(8.36) |
> > > | Default | 84.93(14.17) | 84.82(13.23) |
> > >
> > >
> > > Regarding your question on the sensitivity of hyperparameters for other attacks, we additionally give the result of the sensitivity study under the Trojan-WM attack (due to time constraints, we are unable to finish the study for all attacks).  As can be seen from the following tables, the performance variation trend with regard to the three hyper-parameters is similar and robust across attacks.
> > >
> > >
> > >
> > > | $\alpha=$ | 0.50     | 0.55     | 0.60     | 0.65     | 0.70     | 0.75 | 0.80 | 0.85 | 0.90 | 0.95 |
> > > | --------- | -------- | -------- | -------- | -------- | -------- | -------- | -------- | -------- | -------- | -------- |
> > > | ACC       | 69.89    | 72.36    | 77.82    | 81.43    | 80.72    | 82.48    | 83.43    | 83.37    | 84.82    | 85.19    |
> > > | ASR       | 6.18     | 7.80     | 10.28    | 11.85    | 10.27    | 9.78     | 10.91    | 13.27    | 13.23    | 18.62    |
> > >
> > >
> > > | $\gamma=$ | $10^{-8}$ | $10^{-7}$ | $10^{-6}$ | $10^{-5}$ |
> > > | --------- | --------- | --------- | --------- | --------- |
> > > | ACC       | 85.39     | 84.82     | 84.36     | 74.34     |
> > > | ASR       | 9.39      | 13.23     | 10.77     | 10.13     |
> > >
> > > | $\tau=$ | No Defense | 100   | 500   | 1000  | 1500  | 2000  |
> > > | ------- | ----- | ----- | ----- | ----- | ----- | ----- |
> > > | ACC     | 88.51 | 85.17 | 85.56 | 84.82 | 84.27 | 84.59 |
> > > | ASR     | 99.86 | 8.36  | 10.27 | 13.23 | 16.25 | 24.38 |
> > >
> > > In conclusion, our method is able to deal with different attacks with a default set of hyperparameters and those hyperparameters are indeed robust under various attacks. We hope this address your concerns!

---

> > > > ### Comment · Reviewer_T69H · 2022-08-08
> > > > **Response to response**
> > > >
> > > > I would like to thank the authors for the detailed response and experiments. I guess the additional experiments are conducted on ResNet & CIFAR10. Now I have a question:
> > > >
> > > > In backdoor defense, we usually want to choose a good hyperparameter based on a budget on the drop of ACC. This works fine when the selection of hyperparameters monotonically reduces ACC and ASR at the same time. For example, $\alpha$ in the above experiments. Decreasing $\alpha$ can reduce both ACC and ASR almost monotonically, then we can choose the smallest $\alpha$ that reach the lowest acceptable ACC. Under this choice, ASR is almost the lowest we can reach within the given budget. There are other typical examples like the learning rate in fine-tuning defense, threshold hyperparameters in ANP etc. Hence, the results on $\alpha$ and $\gamma$ is fine to me.
> > > >
> > > > However, I notice that the increase of $\tau$ will reduce the ACC and ASR first, but after some points, the ACC and ASR will increase again ($\tau$=100 for ASR and $\tau$=1500 for ACC). That means it may be hard choose the best $\tau$ within the given budget (the _best_ here means the lowest ASR). For instance, choosing $\tau=1500$ leads to the lowest ACC, but the ASR is not the lowest. I think this property may bring difficulty in tuning the hyperparameters. Although the highest ASR is still acceptable in the given results, it doesn't ensure we can be such lucky under some other cases (other datasets, attacks, architectures etc.). Because we may choose a hyperparameter that have the lowest ACC but very high ASR. Hence, I'm interested in whether the ASR will keep increasing after 2000. This could be very important.

---

> > > > > ### Author Response · Authors · 2022-08-08
> > > > > **Response to Reviewer T69H**
> > > > >
> > > > > Thank you for your reply and questions. Yes, they are conducted on CIFAR10 with ResNet.
> > > > >
> > > > > First, we want to emphasize that in general, the ACC will decrease with the increase of $\tau$ while ASR will first decrease fastly and then slightly increase again. Of course, we will also observe some oscillations, but that’s mainly due to the randomness in optimization. To confirm these trends, we additionally provide the full sensitivity study of $\tau$ on Trojan-WM, Badnets, and $L_0$-inv attacks with $\tau$ ranging from 10 to 3000 (considering the data dimension for CIFAR10, further increasing $\tau$ is the same as no constraint and thus meaningless). We can observe that the results fit our expectations.
> > > > >
> > > > > **Trojan-WM**
> > > > >
> > > > > | $\tau=$ | **No Defense** | **10** | **50** | **100** | **500** | **1000** | **1500** | **2000** | **2500** | **3000** |
> > > > > | --------- | -------- | ------ | ------ | ------- | ------- | -------- | -------- | -------- | -------- | -------- |
> > > > > | ACC       | 88.51    | 86.38  | 85.93  | 85.17   | 85.56   | 84.82    | 84.27    | 84.59    | 84.14    | 83.56    |
> > > > > | ASR       | 99.86    | 51.63  | 9.01   | 8.36    | 10.27   | 13.23    | 16.25    | 24.38    | 22.97    | 25.85    |
> > > > >
> > > > >
> > > > > **Badnets**
> > > > >
> > > > > | $\tau=$ | **No Defense** | **10** | **50** | **100** | **500** | **1000** | **1500** | **2000** | **2500** | **3000** |
> > > > > | --------- | -------- | ------ | ------ | ------- | ------- | -------- | -------- | -------- | -------- | -------- |
> > > > > | ACC       | 87.83    | 86.38  | 85.15  | 84.48   | 84.54   | 84.93    | 83.60    | 83.56    | 82.67    | 82.17    |
> > > > > | ASR       | 97.90    | 24.52  | 11.07  | 12.31   | 13.88   | 14.17    | 13.24    | 12.11    | 13.45    | 13.39    |
> > > > >
> > > > >
> > > > > **L0-inv**
> > > > >
> > > > > | $\tau=$ | **No Defense** | **10** | **50** | **100** | **500** | **1000** | **1500** | **2000** | **2500** | **3000** |
> > > > > | --------- | -------- | ------ | ------ | ------- | ------- | -------- | -------- | -------- | -------- | -------- |
> > > > > | ACC       | 88.23    | 85.81  | 85.71  | 85.60   | 85.39   | 85.76    | 85.18    | 85.54    | 85.47    | 84.34    |
> > > > > | ASR       | 100.0    | 25.97  | 10.13  | 11.06   | 11.52   | 10.26    | 13.69    | 12.77    | 13.85    | 16.29    |
> > > > >
> > > > >
> > > > >
> > > > > The reasons behind this are also easy to understand. In terms of ACC, since we are using a larger  $\tau$, it means that the model is modified (through the masks) to adapt to triggered data points that are further away from the original data distribution. Thus in general, it will hurt the ACC on the original data distribution. In terms of ASR, in an ideal condition, it should be decreasing since the solution space of the optimization problem with larger $\tau$ actually contains the solution of the same problem with a smaller $\tau$. Thus in terms of backdoor removal, it should be at least as good as that of a smaller $\tau$. However, in practice, since we use PGD for solving the inner maximization problem, setting a large $\tau$ will simply make the $L_1$ norm of the recovered trigger very large (similar to adversarial attacks, PGD will make the solution close to the norm ball boundary). Thus the recovered trigger will be further away from the actual trigger and affects the backdoor removal performances.
> > > > >
> > > > > In summary, the ACC will gradually decrease while the ASR will decrease fast at first and then slowly increase. Fortunately, such trends of ACC and ASR actually do not make the selection of $\tau$ too hard. On one hand, as you mentioned, if we pick the lowest acceptable ACC, even in the worst case (with $\tau=3000$) the ASR performance is still acceptable. On the other hand, we argue that selecting a moderate $\tau$ (such as 1000) is a reasonable choice as it can improve upon the worst-case ASR and ACC with little additional risks. Although we do not have prior knowledge of the possible trigger pattern, it is still reasonable to assume that the adversary will tend to make a trigger invisible and small in the norm, otherwise, it would be easy for the human eyes to detect and hard to maintain the clean test accuracy. For example, the $L_1$ norm of Trojan-WM’s actual trigger is around 115 (and the triggers from other attacks also have similar norms), which is way smaller than our choice of $\tau=1000$. In fact, $\tau=1000$ roughly means that we allow the trigger to significantly change over 16x16 pixels in a 32x32 CIFAR image which is already fairly large.
> > > > >
> > > > > We hope the additional results and explanations address your concern.

---

> > > > > > ### Comment · Reviewer_T69H · 2022-08-09
> > > > > > **Response to response**
> > > > > >
> > > > > > I would like to thank the authors for the detailed experiments, and the thoughtful comments, which make me clearer about the method.
> > > > > >
> > > > > > Based on the experimental results for now, I do not find obvious fault of this method. Hence, I raise my score to borderline accept.
> > > > > >
> > > > > > However, I personally think this method is only a heuristic variant of ANP, since the proposed tricks (soft mask, sparse regularization, etc.) do not bring significant insights to the backdoor community. Moreover, although the experimental results look fine on CIFAR10 with ResNet, I still have a little concern about the difficulty of choosing the hyperparameters in other cases (datasets, architectures, attacks). For example, when the number of dimensions of the input images is large, e.g., $3\times224\times224$, we will have $\tau$ ranging from 0 to 150528. I think it would be hard to determine an appropriate $\tau$, not to mention that one has to simultaneously choose $\gamma$ and $\alpha$.
> > > > > >
> > > > > > In summary, judging from the current results, the performance of AWM is acceptable, but the contribution is considered limited. Moreover, it would be much better if the authors can provide richer study over the search space of the hyperparameters.

---

> > > > > > > ### Author Response · Authors · 2022-08-09
> > > > > > > **Thank you and further clarifications on your concerns**
> > > > > > >
> > > > > > > Thanks for your reply and raising your score. Yet we would like to further clarify some points in your last comment and hope to address your concerns on the contribution and hyperparameter search.
> > > > > > >
> > > > > > > In terms of our contributions, we respectively disagree with you. Our method is *NOT* “a heuristic variant of ANP”. The proposed method is for solving a specific problem in ANP that it performs poorly when the available training data size is small. And from our ablation study in Table 3, you can easily observe that the ATR (adversarial trigger recovery) module is the main driving force for the improved performances, not the soft mask, or the sparse regularization. Essentially, we estimate the worst-case perturbation $\Delta$ to the input data for backdoor removal while ANP perturbs the neurons for finding the sensitive ones.  In fact, till now it is hard to understand why there exists such sensitive neurons that directly corresponds to the backdoor (or essentially ANP works aside from the experimental results). Yet our trigger recovery strategy is actually very intuitive and easy to understand: we ask the neural network to adjust the weight masks to make sure that even though the worst-case trigger is applied, the prediction will not be modified, which shared some essence with the adversarial training method in adversarial robustness study. But different from adversarial training, which typically does not work when the finetuing dataset is small, we only adjust the weight masks instead of model parameters itself to make sure that our method works even under one-shot setting. Therefore, we believe that our method opens up new directions for backdoor removal with very limited original data and sheds light on various other tasks that aims to perform fine-tuning with the one-shot setting.
> > > > > > >
> > > > > > > Regarding your question on how to choose the hyperparameters (especially $\tau$) under other dataset/architectures. First, we want to argue that the hyperparameter tuning is a common problem for all current backdoor removal algorithms, not just ours. ANP also need to set the $\epsilon$ values just like $\tau$ in our case. Second, we believe that choosing $\tau$ in our case is actually easier than setting $\epsilon$ in ANP. It is hard to reason how $\epsilon$ (neuron sensitivity) would change when the training data dimension goes from 32x32x3 to 224x224x3. Yet for $\tau$, we can actually decide its value based on the actual image dimension with a fixed ratio. For example, $\tau=1000$ in the 32x32x3 images corresponds to $\tau=49000$ in the 224x224x3 image, which corresponds to fully change the value of a 128x128 region. This actually makes our hyperparameters more interpretable compared with the existing works. Also, please note that in our ablation experiments, we actually have a very wide range of $\tau$ that satisfies our goal of successful backdoor removal, which makes the hyperparameter tuning a less annoying problem.
> > > > > > >
> > > > > > > In summary, we believe that our work has significantly improved upon previous backdoor removal works especially in the one-shot setting and can also inspire many related works on performing fine-tuning with the one-shot setting. We also believe that the hyperparameter tuning (especially $\tau$) is not really a problem and actually our hyperparameters are more interpretable than the existing works. We hope this further address your remaining concerns and we will be really appreciated if you can further consider our response here in the discussion phase.

---

### Official Review · Reviewer_kGgj · 2022-07-09

**Rating:** 5
**Confidence:** 4
**Soundness:** 3 good
**Presentation:** 3 good
**Contribution:** 3 good

**Summary:**

This paper tries to improve the recent proposal of ANP in the small training data range, by using an "optimization" based procedure to find a soft-mask, in order to remove the backdoor trigger pattern.

The approach is well motivated, and well formulated (4.1--4.4)

Experiments show the effectiveness of the method, especially in the one-shot case

**Questions:**

Are there possibilities for adaptive attacks if the adversary knows the optimization procedure for removing the backdoor pattern? Is this a sensible question to ask?

**Limitations:**

No limitation identified

**Strengths And Weaknesses:**

Strengths: Well motivated approach. The presentation is very clear and strong experiment results are shown

Weakness: I'm not sure whether this has been studied, but if the adversary is aware of the optimization procedure for removing the backdoor pattern, would he be able to leverage that to counter the defense?

---

> ### Author Response · Authors · 2022-08-02
> **Response to Reviewer kGgj**
>
> Thanks for your question!
>
> Q1: Possible design of adaptive attack methods if the adversary knows the optimization procedure.
>
> A1: This is an interesting question. To the best of our knowledge, we have not seen adaptive attacks studying the case with prior knowledge of the backdoor removal procedure. Some principles in developing backdoor attacks, such as making triggers invisible or natural, are data-driven and do not direct a specific backdoor removal optimization procedure. Potentially, the corresponding adaptive attack can be hard on the device since the current backdoor removal techniques including ours, have already involved complicated bi-level optimization.
>
> Instead, we design a simple method targeting our backdoor trigger reconstruction mechanism: we only estimate one universal trigger $\Delta$ in every iteration. Now suppose the attacker knows our design and decides to inject multiple backdoor patterns (e.g., 3 trigger patterns) into the model. They would hope our design could only remove one of them and thus fail on the other triggers.
>
> The following table summarizes the result of defending against multiple backdoor triggers on CIFAR10. We first train a multiple backdoored model using three different types of triggers: Badnets trigger, Trojan-WM trigger, and $L_2$-inv trigger. The overall poison rate is set as $5\\%$. Then we apply our AWM to remove the backdoors. ASR (all) represents the attack success rate as long as any one of three triggers fooled the model. ASR1, ASR2, and ASR3 represent the attack success rate of each of the three triggers. ACC is the test accuracy on the clean test set. The results show that adding multiple triggers still cannot penetrate our AWM defense even with limited resources.
>
> | ShuffleNet | ASR (all)  | ASR1 | ASR2 | ASR3 | ACC  |
> | -------- | ----- | ----- | ----- | ----- | ----- |
> | Original | 99.83 | 95.80 | 99.52 | 99.15 | 84.86 |
> | AWM(500 images) | 10.38 | 9.54  | 8.19  | 13.16 | 77.02 |
> | AWM(one-shot) | 8.47  | 13.54 | 10.34 | 16.21 | 71.83 |
>
> We conjecture that the reason for this result is that although multiple triggers are involved, our algorithm will still try to identify the most likely triggers in each iteration. Thus when the first (the most prominent) ‘trigger’ is removed, the algorithm will automatically target the next likely ‘trigger’.
>
> We believe that such backdoor removal strategies can be hard to penetrate. Successful attacks may need to leverage tri-level optimization problems, which are notoriously hard to solve, or aim to make the removal strategy impractical by lowering its natural accuracy, which currently has no concrete solutions. We will leave this problem as one of our future work directions.

---

> ### Author Response · Authors · 2022-08-09
> **A friendly reminder of the rebuttal conclusion**
>
> We would like to thank you for your comments. We have responded to your question and hope it could help address your concern. In addition, we are more than happy to discuss and address any further questions before the conclusion of the rebuttal period.

---

### Official Review · Reviewer_uwMb · 2022-07-11

**Rating:** 6
**Confidence:** 4
**Soundness:** 3 good
**Presentation:** 3 good
**Contribution:** 2 fair

**Summary:**

This paper improves upon the previous backdoor defense method ANP, by replacing the binary neuron pruning in ANP with a more flexible soft weight masking method. Experimental results show performance improvements over ANP.

**Questions:**

Please see above.

**Limitations:**

Please see above

**Strengths And Weaknesses:**

Strengths:
1. All three technical improvements made over ANP are intuitively correct. And the results validate their effectiveness.
2. The paper is well-written and easy to follow.

Weakness:
1. The technique novelty is limited.
The proposed method is a minor modification over ANP. The main difference is that it replaces hard-threshold binary masking in ANP with a more flexible soft weight masking/reweighting. Comparing Eq 3.2 (ie ANP) with Eq 4.2 (ie the preliminary proposed method), we can see the difference is only in the range of the mask: ANP require the mask to be binary while AWM (the proposed method) allows it to be continuous values in range [0,1]. Although AWM further used two more modification (sparsity on recovered backdoor pattern \delta and sparsity on the mask m), ablation study results in Table 3 show they don't bring considerable improvements: the gain is mainly from the use of softmasks. I would say the proposed method works as people would expect, but only as a minor modification over ANP. Also, the sparsity regularization on recovered backdoor pattern (Eq 4.2) is also hardly a novel contribution. It has been used in Neural Cleanse [1] and many follow-up works such as [2].

[1] Neural Cleanse: Identifying and Mitigating Backdoor Attacks in Neural Networks. S&P, 2019.
[2] Better Trigger Inversion Optimization in Backdoor Scanning. CVPR, 2022.

2. The reported results on the baseline method IBAU is too low.
In table 1, the authors reported IBAU has over 30% attack success rate (ASR) against Trojan-WM when 100 clean samples are available. However, in the original IBAU paper, in their Table 5, under the same setting, the ASR is as low as 4%. Can you please explain why there is such large gap, or point out what difference in the setting I have overlooked? Thanks!

3. This paper highlights its performance under one-shot setting. It is indeed advantage if the backdoor defense method requires less clean samples to fix the poisoned model. However, why do we need to peruse the extreme case of one-shot (ie only one clean sample is available for defense)? I think it practical for the defender to have a small amount (eg some hundreds) of clean data.

---

> ### Author Response · Authors · 2022-08-02
> **Response to Reviewer uwMb**
>
> Thank you for your comments!
>
> Q1: The difference between ANP and AWM. The novelty of the algorithm.
>
> A1: We respectively disagree with your opinion on the novelty of our work. Please note that the mentioned binary/soft masking, as well as the sparsify on $m$, are just minor tricks used in our method, not our major contributions. The formulations of ANP and AWM have two other major differences: (1) we put masks on the model weights while ANP focuses on masking the neurons (as shown in Section 3, this gives us advantages in feasibly saving benign information especially when the network is small); (2) we estimate the worst-case perturbation $\Delta$ to the input data for backdoor removal while ANP perturbs the neurons for finding the sensitive ones (which represents two totally different strategies for backdoor removal and explains why they adopt binary mask while we adopt soft mask). The above two steps, noted as SWM and ATR, both contribute to the performance improvements as can be seen in Table 3. Although the sparsity constraints on the weight masks do not bring significant improvements, we additionally show and discuss the impacts of different forms of sparsity constraints in Table 4. Nevertheless, we will cite and discuss the two sparsity regularization strategies (update and cite in the paper revision)
>
> Q2: Explain the performance gap with the original IBAU paper.
>
> A2: We are sorry for the confusion here. Please note that in our paper, we consider a different poison rate from the original IBAU paper, which leads to the performance difference. In Table 5 of IBAU, the poison rate is set as 0.20, while we follow ANP and adopt a poison rate of 0.05 in our paper. We believe that the poison rate of 0.20 is too large and impractical in real-world scenarios and a lower poison rate will further demonstrate the power of the proposed backdoor removal method.
>
> Q3: The reason why we need the one-shot unlearning setting.
>
> A3: We do agree that the one-shot case is an extreme setting. In practice, we may have a small amount of training data available. Yet we want to argue that our method not only works much better under the one-shot case but also works better or at least comparatively well in other cases. As can be seen from Table 1 in the paper and Table 3 in the appendix, in scenarios with a few hundred clean data, AWM still largely outperforms other backdoor removal baselines. Moreover, in some cases, the defender may have no clue what is the training data at all. Under such settings, it would be wonderful if the model owner could also remove the backdoors in a data-free way, which is our future aim. Studying the backdoor defense methods under the one-shot case brings us closer to that goal.

---

> > ### Comment · Reviewer_uwMb · 2022-08-03
> > **Thank you and the score is raised.**
> >
> > My concerns are solved. I raised my score to weak accept. Thank you!

---

> > > ### Author Response · Authors · 2022-08-03
> > > **Thank you**
> > >
> > > We are glad that our responses have addressed your concerns. Thank you for raising the score.

---

### Meta-Review · Area_Chair_Gaoi · 2022-08-31

**Recommendation:** Accept
**Confidence:** Less certain

**Metareview:**

The paper presents a method for defending deep neural networks against backdoor attacks, i.e., attacks that inject “triggered” samples into the training set. The method can be seen as an improvement on Adversarial Neuron Pruning (ANP) that uses (i) soft weight masking (SWM), (ii) adversarial trigger recovery (ATR) and (iii)  sparsity regularization (SR). The main focus of the paper is in the low-data regime, especially in the one-shot setting and when the network size is small.

The authors have clarified the novelty of the approach wrt to ANP and have provided additional experiments addressing some of the reviewers' concerns. In view of this, some of the reviewers raised their scores. However, there are still concerns regarding the novelty of the method and the difficulty of setting hyperparameters. The empirical results seem solid, however.


**Award:**

No

---

### Decision · Program_Chairs · 2022-09-14

Accept